# High-Gravity Fermentation for Bioethanol Production from Industrial Spent Black Cherry Brine Supplemented with Whey

**Javier Ricardo Gómez Cardozo** , **Jean-Baptiste Beigbeder**, **Julia Maria de Madeiros Dantas** and **Jean-Michel Lavoie** *

Biomass Technology Laboratory, Department of Chemical Engineering and Biotechnology Engineering, Université de Sherbrooke, 2500 Boul. de l'Université, Sherbrooke, QC J1K 2R1, Canada

* Correspondence: jean-michel.lavoie2@usherbrooke.ca

**Abstract:** By-products from different industries could represent an available source of carbon and nitrogen which could be used for bioethanol production using conventional *Saccharomyces cerevisiae* yeast. Spent cherry brine and whey are acid food by-products which have a high organic matter content and toxic compounds, and their discharges represent significant environmental and economic challenges. In this study, different combinations of urea, yeast concentrations, and whey as a nutrient source were tested for bioethanol production scale-up using 96-well microplates as well as 7.5 L to 100 L bioreactors. For bioethanol production in vials, the addition of urea allowed increasing the bioethanol yield by about 10%. Bioethanol production in the 7.5 L and 100 L bioreactors was 73.2 g·L$^{-1}$ and 103.5 g·L$^{-1}$ with a sugar consumption of 81.5% and 94.8%, respectively, using spent cherry brine diluted into whey (200 g·L$^{-1}$ of total sugars) supplemented with 0.5 g·L$^{-1}$ urea and 0.5 g·L$^{-1}$ yeast at 30 °C and a pH of 5.0 after 96 h of fermentation for both systems. The results allow these by-products to be considered low-economic-value alternatives for fuel- or food-grade bioethanol production.

**Keywords:** bioethanol; fermentation; yeast; cherry brine; whey; industrial by-products



## 1. Introduction

The global biofuel market is led by biodiesel and bioethanol production, in which the production of the latter corresponds to about 74% of the biofuel industry [1]. Bioethanol is a commodity used in many industries, such as medicine, pharmacy and biofuel without forgetting as an additive for gasoline to cut down the emissions of pollutants. In this sense, this compound has been important in the race against reducing our dependence on fossil fuels and greenhouse gas emissions. Bioethanol is obtained via the fermentation of sugars present in several sources, such as sugarcane, corn, sugar beet, wheat and sweet sorghum, as well as agricultural, industrial and forest by-products [2,3].

The valorization of industrial by-products for bioethanol production has gained more interest at the industrial level to reduce overall production process costs. Among them, molasses [4], whey [5], hydrolyzed lignocellulosic materials [6] and by-products from different industries [7] have been evaluated to obtain this commodity.

Black cherry (*Prunus serotina)* fruits are used for human consumption in many forms, such as fresh, canned, juiced and dried, but also as a flavoring for soft drinks and in jams and jellies. Nonetheless, black cherry brine is an acid solution produced as a preservation medium for cherry fruits before being used in food products, especially in the ice cream industry. Its composition depends on the formulation of the producers and cherry type. Typically, cherries, water, sugars (4.0–11.0% *w/v*) and starches are present in this medium, as well as natural flavors, colorants and preservatives, such as citric acid, sodium benzoate, potassium sorbate, calcium chloride and sulfites, used to avoid microbial contamination [8]. Once the fruits are removed, the solution remains as a by-product that must be treated

before its disposal due to its high biological (DBO) and chemical oxygen demand (DCO), toxicity and acidity.

Whey, on the other hand, is the main by-product of the cheese industry. It is estimated that, for every 1 kg of cheese produced, 9 kg of whey is generated [9]. This makes it an abundant by-product, containing an important organic (lactose, 3.8–5.0% *w/v*, and proteins, 0.8–1.0% *w/v*) and inorganic (0.5–0.7% *w/v*) load [10]. However, it is estimated that about 50% of the total whey production (189 million tons/year) is considered as a waste and is disposed of without any treatment [11]. Therefore, an alternative would be to use it as a supplement for fermentation processes, and although it is true that yeasts do not usually metabolize lactose, the nutrients present in whey can be used in the enzymatic processes of these microorganisms.

This study focuses on the recovery of these two industrial residues (spent cherry brine, hereafter simply called cherry juice, and whey from a local company, located in the Quebec Province, Canada). Fermentations were carried out using cherry juice as a carbon source and whey as a nutritional supplement, aiming to improve the sugar consumption present in cherry juice and increase the yield in ethanol production. For this, growth tests were carried out in microplates, after which the process was scaled up to bioreactors with capacities of 7.5 L and 100 L, respectively.

To our knowledge, this is the first time that whey and cherry juice were combined for fermentation, which is an alternative use of such by-products for the sustainable production of bioethanol.

## 2. Materials and Methods

### 2.1. Cherry Juice and Whey Characterization

Both cherry juice and whey were collected from a local sugar refinery located in Coaticook (Québec, Canada) and stored at 4 °C and −20 °C, respectively, until further use. The cherry juice (Black Cheery Halves, $_{MD}$Cibona$^{TM}$, Lot# C21236) was the liquid part used to preserve the black cherry fruit before being used in various Laiterie de Coaticook Ltée products. This preserving liquid was mainly composed of sugar, water, citric acid, sodium benzoate, $CaCl_2$, natural and/or artificial flavor, color and sulfites, according to the producer. The whey was generated from cheese production by the same company. The cherry juice and whey were initially characterized at the Biomass Technology Laboratory analytical facilities to identify the concentrations of compounds such as sugars and organic acids, as well as for an elemental analysis to quantify the Carbon (C), Hydrogen (H), Nitrogen (N) and Oxygen (O) content (see Table 1).

### 2.2. Yeast Inoculum, Culture Media and Growth Curves in Microplates

Yeast inoculum was prepared by rehydrating *Saccharomyces cerevisiae* dry yeast cells (DistilaMax® HT, Lallemand Biofuels & Distilled Spirits, Canada) with tap water for 15 min at 30 °C and 140 rpm using a shaking incubator (Infors-HT Inc., Bottmingen, Switzerland).

The composition of the media used to evaluate yeast growth was cherry juice at 60 g·L$^{-1}$ of total fermentable sugars (6.0°Brix) diluted in tap water or whey, supplemented with urea, Crystal Reagent, ACS, 99.0–100.5%, Anachemia, Mississauga, ON, Canada, Inc. (0.0 g L$^{-1}$, 0.5 g·L$^{-1}$, and 1.0 g·L$^{-1}$), at two initial yeast concentrations (0.5 g·L$^{-1}$ and 1.0 g·L$^{-1}$). Different yeast suspensions were prepared to obtain the desired initial concentration for each experiment. The initial pH of the medium was adjusted to 5.0 with NaOH 1.0 N. Fermentations were performed by adding 200 μL of the culture medium inoculated according to the initial yeast concentration in a 96-well microplate, and each condition was evaluated in triplicate. The plates were then shaken in a thermostat microplate reader (BioTek EPOCH2) at 30 °C for 49 cycles (about 24 h in total) of double orbital shaking (1 min at 282 rpm) followed by optical density measurements at 600 nm (OD$_{600}$). The data collected were converted to biomass using a calibration curve (see Figure S1). All the experiments were conducted in triplicate, and the results represent the Mean (M) ± Standard Error (STD).

**Table 1.** Characterization of cherry juice and whey used in this study.

| Parameter | Cherry Juice | Whey |
|---|---|---|
| pH | 3.3 | 4.0 |
| Density (g·mL$^{-1}$) | 1.2 ± 0.1 | 1.1 ± 0.1 |
| Humidity (%wt) | 61.6 ± 0.2 | 93.7 ± 0.0 |
| Total Solid (%wt) | 38.4 ± 0.2 | 6.3 ± 0.0 |
| Ashes (%wt) | 0.1 ± 0.0 | 0.6 ± 0.0 |
| Total Protein (g·L$^{-1}$) | - | 0.5 ± 0.1 |
| **Sugars (g·L$^{-1}$)** | | |
| Sucrose | 248.0 ± 0.8 | - |
| Lactose | - | 41.9 ± 0.1 |
| Glucose | 105.6 ± 0.3 | - |
| Fructose | 99.4 ± 1.1 | - |
| Total sugars | 453.0 ± 2.2 | 41.9 ± 0.1 |
| **Other Compounds (g·L$^{-1}$)** | | |
| Lactic Acid | - | 33.1 ± 0.5 |
| Citric Acid | 3.9 ± 0.1 | - |
| Sodium Benzoate | 1.2 ± 0.1 | - |
| **Elemental Analysis (%wt)** | | |
| Carbon | 17.0 ± 0.0 | 2.4 ± 0.5 |
| Hydrogen | 2.6 ± 0.2 | 0.3 ± 0.1 |
| Nitrogen | <LOD | 0.1 ± 0.0 |
| Oxygen | 22.0 ± 1.1 | 2.2 ± 0.3 |

LOD: Limit of detection.

### 2.3. Culture Media and Fermentation in Vials

The culture media employed in all fermentations was composed of cherry juice that was added to whey to reach 200 g·L$^{-1}$ (around 20 °Bx) of total fermentable sugars (sucrose, glucose and fructose). The initial and final sugar concentrations of the mixture of cherry juice and whey were measured using a DIONEX ICS-500+ ion chromatography system (see analytical procedures). The initial pH of the medium was adjusted to 5.0 with NaOH 1.0 N. Carbon dioxide ($CO_2$) production indicated the metabolism of the sugars present in the medium. To monitor fermentations, a gravimetric method was used to measure the amount of gas released according to the weight difference, using an accurate balance (0.001 g). The final ethanol concentration was quantified via HPLC (see analytical procedures).

The fermentations were carried out in 50 mL vials with rubber septum stoppers and aluminum rings, with a working volume of 30 mL using the culture media described above. The initial yeast concentration used was 0.5 g·L$^{-1}$. Urea was used as a nitrogen source at two concentrations (0.0 g·L$^{-1}$ and 0.5 g·L$^{-1}$) to evaluate the effects on the ethanol yield. Fermentation ran under anaerobic conditions (by flushing the vials with $N_{2(g)}$ for 1.0 min) for 170 h at 30 °C and 140 rpm.

### 2.4. Bioethanol Production Using Cherry Juice and Whey in Bioreactors

Bioethanol production in the bioreactors was evaluated using the best conditions from screening. Based on the experimental results obtained in 50 mL vials, yeast and total reducing sugar concentrations were used to keep constant fermentation media conditions in a 7.5 L and in a 100 L batch bioreactor with a working volume of 5.0 L and 50.0 L, respectively. The 7.5 L batch bioreactor (Infors-HT Inc., Bottmingen, Switzerland) was operated at 30 °C and was stirred at 200 rpm using 2 Rushton turbine impellers. The 100 L bioreactor, a 0.5 m tall vertical cylinder made from stainless with 0.5 m of internal diameter, equipped with a 2000 W heating element to control the internal temperature of the system and an agitation system consisting of a 2-blade inclined impeller powered by a 12 V DC electric motor, was operated at 30 °C and 70 rpm. Samples were taken in triplicate, and the data reported were the M ± STD of the samples analyzed from each batch.

### 2.5. Analytical Procedures

Brix degrees (°Brix), or the total soluble solid content of cherry juice media, were measured using a digital pocket refractometer (ATAGO, PAL-BX/RI, Bellevue, WA, USA). Samples were tested in triplicate, and the data reported were the Mean (M) ± Standard Error (STD) of each one.

The determination of moisture and dry matter was carried out according to the mass loss, for which around 5.0 g of cherry juice and whey were weighed in previously weighed aluminum dishes, and the samples were dried at 105 °C in an oven (Fisher Sci 100L, Pittsburg, PA, USA) for 12 h or until a constant weight was reached. The dry matter obtained after moisture determination was used to quantify the CHNS/O content with an elemental analyzer (Flash 2000 OEA, Thermo Fisher Scientific, Toronto, ON, Canada), as well as the ash content, which was determined according to the NREL/TP-510-42622 method [12]. Finally, the total protein content was determined following the method described by Bradford [13]. In all cases, samples were tested in triplicate, and the data reported were the M ± STD for each parameter.

High-performance liquid chromatography (HPLC) was used to quantify ethanol and lactic acid. Samples were diluted, filtered and injected into the chromatographic system (Agilent 1100 series equipped with a G1362A Refractive Index Detector) (Agilent Technologies Inc., Colorado Springs, CO, USA). The system was operated at 40 °C with an isocratic elution method (2.5 mM). The HPLC set-up also had a G1322A Degasser and a G1311A Quaternary Pump. A G1313A Autosampler injected 40 µL of the sample, and the column used was ROA-Organic Acid H$^+$ (8%) at 65 °C. The elution was performed at a constant flow of 0.600 mL·min$^{-1}$ of a 2.5 mM H$_2$SO$_4$ solution. A calibration curve from 10 ppm to 1000 ppm was performed using the following standards: L-lactic 99% (Alfa Aesar, Tewkbury, MA, USA), ethanol 99% (Sigma Aldrich, Oakville, ON, Canada) and glycerol 99% (Sigma Aldrich, Oakville, ON, Canada).

The DIONEX ICS-500+ ion chromatography system allowed the quantification of the total fermentable sugars (sucrose, glucose and fructose), citric acid and sodium benzoate. The system was equipped with a KOH eluent generator to provide a proper eluent concentration. A 200 mM KOH post-injection with a Dionex GP 50 gradient pump was implemented to ensure signal stability. A Dionex CarboPac Sa10-4 µM column was used. The oven was set to 45 °C, and the electrochemical detector was at 30 °C. The injection volume was 0.4 µL, and elution was made with an aqueous solution of KOH at 1.25 mL·min$^{-1}$ at the following concentrations: 1 mM for 12 min, 10 mM for 5 min and 1 mM for 1 min. For sugars, the calibration curve involved a concentration of standards varying from 10 ppm to 1000 ppm and involved the following standards: sucrose 99%, glucose 99% and fructose 99%, which were all purchased from Sigma-Aldrich. For citric acid and sodium benzoate, the chromatograms of the standard samples, citric acid 99% (Sigma-Aldrich) and sodium benzoate 99% (Sigma-Aldrich) with a concentration of 1000 ppm, were taken and compared with the chromatograms of the raw cherry juice and whey samples.

### 2.6. Ethanol Yield, Sugar and Productivity

The fermentation performance was evaluated using the ethanol yield ($Y_{EtOH}$) based on the theoretical ethanol production and using the sugar uptake according to Equations (1) and (2). Ethanol productivity was calculated as the grams of ethanol produced per day per liter of liquid volume of the bioreactor.

$$Y_{EtOH}(\%) = \frac{EtOH}{0.51 \cdot S_0} \times 100 \tag{1}$$

$$Sugar\ Consumption(\%) = \frac{S_0 - S_f}{S_0} \times 100 \tag{2}$$

where:

*EtOH:* Ethanol concentration (g·L$^{-1}$);

$0.51 \cdot S_0$: Theoretical ethanol concentration from hexoses ($g \cdot L^{-1}$);

$S_0$: Initial fermentable sugars (Sucrose as C6, Glucose and Fructose ($g \cdot L^{-1}$);

$S_f$: Final fermentable sugars (Sucrose as C6, Glucose and Fructose) ($g \cdot L^{-1}$).

## 3. Results and Discussion

### 3.1. Yeast Growth in Cherry Juice Diluted into Whey

In order to evaluate the yeast growth using cherry juice as a substrate, some tests were carried out to determine the possible toxic effects of the medium on *Saccharomyces cerevisiae* in a microplate system. In addition, the media were supplemented with different urea concentrations because it was initially known that cherry juice has no nitrogen sources, as shown in Table 1. Moreover, tests were carried out by diluting the cherry juice in whey with and without the addition of a nitrogen source.

After 12 h of fermentation, it was observed that the media that used whey as a diluent reached the maximum values for the conditions evaluated, whereas in the media in which the diluent was tap water, slower growth was observed (Figure 1). Increasing the initial yeast load from 0.5 to 1.0 $g \cdot L^{-1}$ improved the growth kinetics of the cherry juice media diluted with water and supplemented with urea. In addition, growth inhibition was observed in the media in which water was used as diluent and no nitrogen source was added, due to the lack of nutrients that are not supplemented only with the addition of urea. It is known that yeasts in their fermentation processes use nitrogen for their growth [14], and the presence of citric acid and sodium benzoate in cherry juice could inhibit its growth [15] even when the medium is supplemented with urea. Finally, the addition of urea does not significantly affect yeast growth in whey-supplemented media. For instance, at 0.5 $g \cdot L^{-1}$ of yeast, CJWU0.0 followed the same trend as CJWU0.5 and CJWU1.0, which means that whey provided enough nutrients to the media without adding urea. However, when using 1.0 $g \cdot L^{-1}$ of yeast, it was observed that the inhibition was lower, and the growth was similar for all the combinations tested, except for CJU0.0, in which strong growth inhibition was observed due to the high toxicity of the medium.

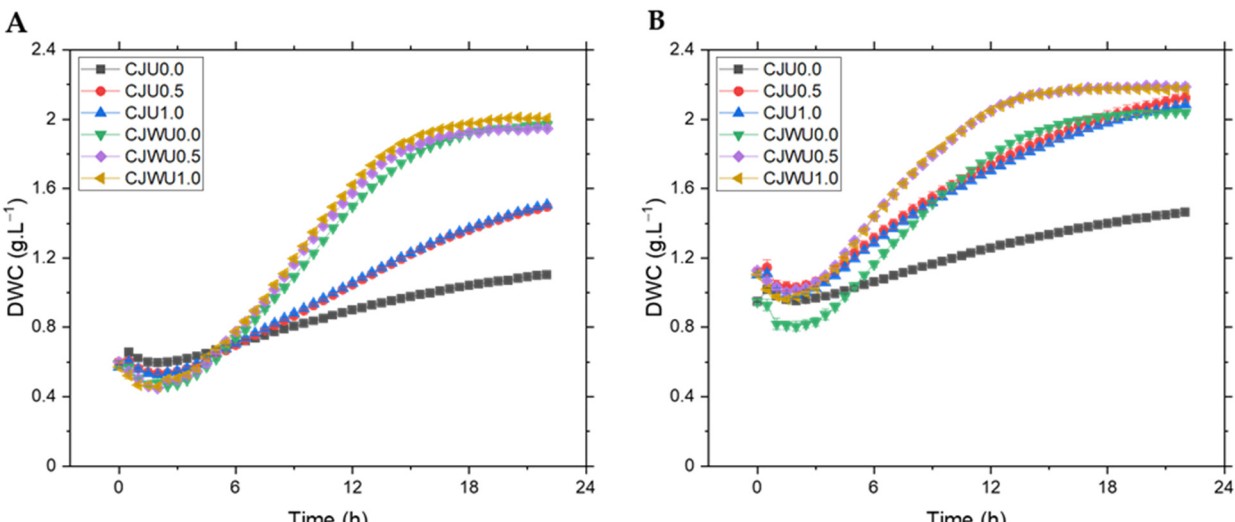

**Figure 1.** Yeast growth curves in the different fermentation media tested: Cherry Juice diluted in Tap Water (CJ) at 6.0°Brix; Cherry Juice diluted in Whey (CJW) at 6.0°Brix; Urea (0.0 $g \cdot L^{-1}$, 0.5 $g \cdot L^{-1}$ and 1.0 $g \cdot L^{-1}$) (U0.0, U0.5 and U1.0) and initial yeast concentrations (**A**) 0.5 $g \cdot L^{-1}$ and (**B**) 1.0 $g \cdot L^{-1}$. All the tests were performed at pH 5.0 (e.g., CJU0.5: Cherry Juice diluted in Tap Water supplemented with 0.5 $g \cdot L^{-1}$ urea. CJWU0.5: Cherry Juice diluted in Whey supplemented with 0.5 $g \cdot L^{-1}$ Urea).

The results also showed that whey contained the nutritional value necessary for the yeast uptake of sugars present in the cherry juice for its growth despite the yeast-inhibiting compounds, such as citric acid and sodium benzoate, which were present in the juice

(Table 1). For example, citric acid has been reported to be a yeast growth inhibitor, reducing its growth rate between 64% and 88% when compared to media without the addition of this compound [15]. On the other hand, a study conducted by Yardimci et al. reported that the yeast growth decreased from 84% to 35% when the concentration of sodium benzoate was duplicated from 25 to 50 mM in YPDA media [16]. However, in our study, in the medium that was diluted with whey, lag, exponential and stationary stages were observed, describing the typical microbial growth in such processes. Studies on the valorization of whey have demonstrated the mineral and protein content of this dairy industry by-product [10,11], which explains why whey-containing media were less toxic than those that only used urea.

The yeast used in this study had a strong tolerance to high-gravity media, and the fact that diluting the cherry juice in whey decreased the inhibitory effect of the toxic compounds it contained led to the use of high-gravity media using these by-products at a weight ratio of cherry juice/whey of 1.11 to reach approximately 200 $g \cdot L^{-1}$ (20°Brix) of total fermentable sugars and to evaluate ethanol production, as described in the following sections.

### 3.2. Fermentation and $CO_2$ Production

The growth kinetics results obtained in the last section allowed us to establish the initial fermentation conditions in order to prepare the yeast inoculum for a series of new fermentation trials. In this context, the initial concentration of sugar was fixed to 20°Brix to simulate industrial fementation and to maximize the ethanol productivity. To delineate the effects of urea in yeast growth and ethanol production, the $CO_2$ production and ethanol yield were measured, as discussed in the following section.

$CO_2$ production shows the fermentative activity of yeast in different media, which can be directly correlated with ethanol production, as presented in Section 2.3. In this study, the results showed a direct relationship between $CO_2$ production and ethanol production, demonstrating the metabolic activity of the yeast on the sugars present in the media (Figure 2). The cherry juice incubated in the absence of nutrients and/or whey (CJ20U0.0) generated very limited production of $CO_2$ ($20.2 \pm 1.7$ $g \cdot L^{-1}$) and ethanol yield ($18.6 \pm 1.8\%$). This behavior was caused by yeast-inhibiting compounds present in the cherry juice along with the production of ethanol and glycerol, two other compounds that are toxic for yeast, which limited the consumption of fermentable sugars present in the medium, such as fructose, decreasing the yields in the fermentation process. However, the supplementation of 0.5 $g \cdot L^{-1}$ of urea in the same media (CJ20U0.5) showed significant effects on the $CO_2$ production kinetics, reaching up to $72.2 \pm 0.8$ $g \cdot L^{-1}$ of $CO_2$.

As expected, the cherry juice diluted in whey presented high fermentation responses. The addition of a nitrogen source in whey-containing media increased yields by approximately 11%, from $77.3 \pm 1.5\%$ in the media supplemented with whey (CJW20U0.0) to $87.1 \pm 4.2\%$ in the media supplemented with whey plus 0.5 $g \cdot L^{-1}$ of urea (CJW20U0.5). The addition of urea in high-gravity media such as 20.8°Brix has been reported to improve ethanol production yields [17] using higher concentrations of urea (2.75 $g \cdot L^{-1}$). However, a high urera content is not recommended because it might favor the production of carcinogenic compounds, such as ethyl carbamate [18]. Considering that, even if the whey provides certain minerals and nutrients to enhance yeast growth (see Table S1), the addition of a nitrogen source favors yeast metabolism for ethanol production. Finally, it should be noted that, although the fermentation processes in the whey-diluted media were carried out up to 170 h, it was observed that, after 120 h, the process reached its maximum value.

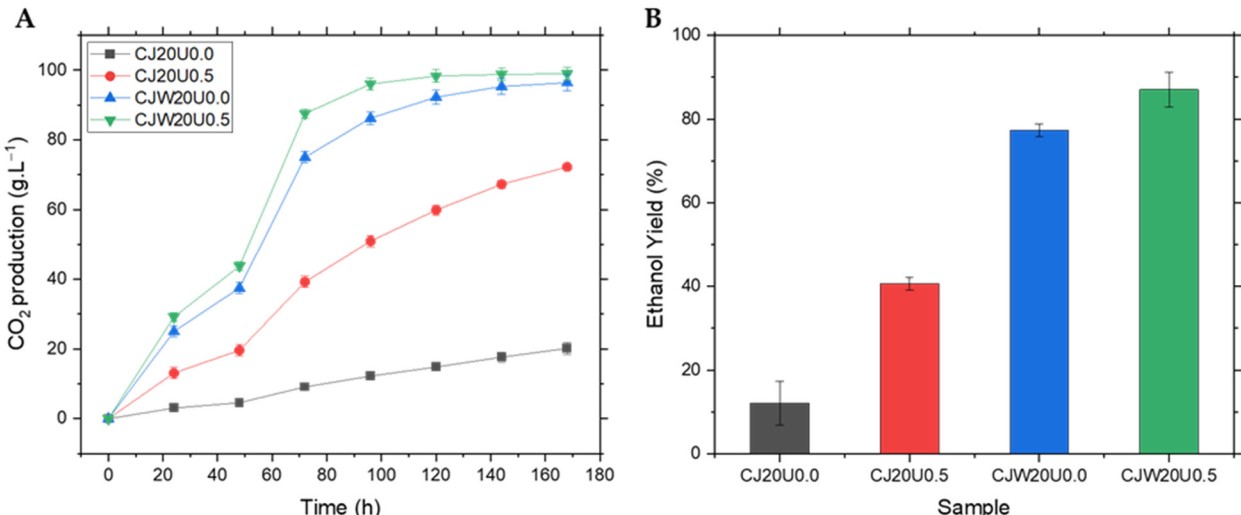

**Figure 2.** (**A**) $CO_2$ production in the different fermentation media tested: Cherry Juice diluted in Tap Water (CJ20) at 20°Brix; Cherry Juice diluted in Whey (CJW20) at 20°Brix; Urea (0.0 g·L$^{-1}$, and 0.5 g·L$^{-1}$) (U0.0, and U0.5) and initial yeast concentration of 0.5 g·L$^{-1}$ at 30 °C and pH 5.0. (**B**) Ethanol yield in the different fermentation media tested at 170 h of incubation (e.g., CJ20U0.5: Cherry Juice diluted in Tap Water at 20°Brix supplemented with 0.5 g·L$^{-1}$ Urea. CJ20WU0.5: Cherry Juice diluted in Whey at 20°Brix supplemented with 0.5 g·L$^{-1}$ urea).

In this section, the results we found are comparable to those presented by Park and Bakalinsky [8] who reported an ethanol yield of 92%. However, they used a spent cherry brine pretreated with alkaline and acidic processes to detoxify it prior to supplementing it with different nitrogen sources, such as urea and diammonium phosphate (DAP). In that sense, this would result in more stages, which would require more energy and increase the costs of the fermentation process.

### 3.3. Bioethanol Production Using Bioreactors

Once the laboratory-scale fermentations were completed, the conditions were used for scaling-up ethanol production using 7.5 L and 100 L bioreactors. The yeast growth took place inside the bioreactors under anaerobic conditions. Liquid samples were taken every 24 h to evaluate the sugar consumption and ethanol production for 120 h of incubation. Samples were stored at −20 °C until analysis.

After 24 h of fermentation, sucrose was quickly hydrolyzed into glucose and fructose molecules, and ethanol production started in both reactors (Figure 3). Glucose was fully consumed after 72 h in both systems, and fructose degradation began after 24 h. The maximum ethanol production was obtained after 96 h of fermentation, reaching values up to 73.2 ± 6.0 g·L$^{-1}$ and 103.5 ± 2.3 g·L$^{-1}$ in the 7.5 L and 100 L bioreactors, respectively. However, the sugar profiles were different due to the operating conditions used in each case because these bioreactors did not have the same geometrical configuration and operating capacity. Nonetheless, scaling-up was performed based just on the optimal conditions for ethanol production from the yeast used in this study (pH and temperature). This had a major impact on the yields because, after 72 h in 7.5 L bioreactor, the yeast was not able to degrade fructose remaining in the media. Factors such as agitation [19] and ethanol concentration can affect the sugar consumption. Thus, a comparison could be performed, as shown in Table 2.

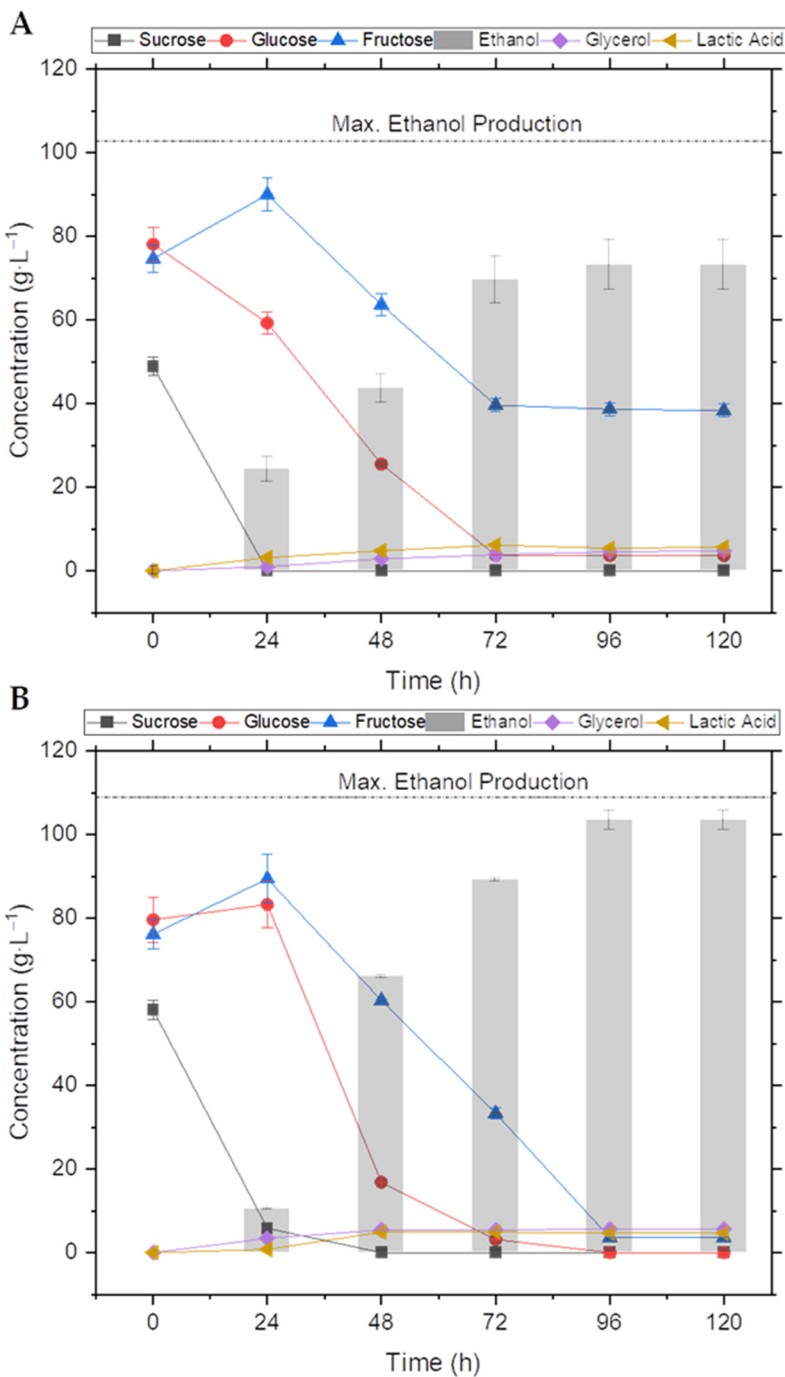

**Figure 3.** Variations in sugars, ethanol, glycerol and lactic acid concentration during fermentation scale-up trials for volumes of 7.5 L (**A**) and 100 L (**B**) in a batch reactor using an initial yeast concentration of 0.5 g·L$^{-1}$, an initial urea concentration of 0.5 g·L$^{-1}$, cherry juice diluted into whey at 200 g·L$^{-1}$ (20°Brix) at 30 °C, a pH of 5.0, and 200 rpm and 70 rpm, respectively.

This difference in system behavior has been reported in a previous study using sugar beet molasses, in which using the same bioreactor systems reached different ethanol yields [20]. From the literature, it is known that, when these scaling processes are performed, it is important to maintain certain parameters, both dimensional configurations [22] and mass transfer, such as power input [21], to minimize variations in the microenvironments where biological processes are performed. Nonetheless, the ethanol concentration and ethanol yield are comparable with those found in the literature, as we can see in Table 2. On the other hand, the selected operational conditions allowed the yeast to uptake most of

the sugar available in media composed by cherry juice diluted into whey. Likewise, the concentration of by-products such as glycerol and lactic acid was similar for both scales, indicating that the stress generated by the scale changes did not affect the yeast metabolism, always favoring ethanol production.

**Table 2.** Comparison of fermentation performance and by-products measured for different substrates: Cherry Juice diluted into Whey (200 g·L$^{-1}$ of total sugars) (this study). Sugar Beet Molasses (170 g·L$^{-1}$ of total sugar) [20] *. Hydrolyzed Raw Cassava Starch (200 g·L$^{-1}$) [21] **.

| Substrates/Parameters | Cherry Juice/Whey | | Sugar Beet Molasses * | | Hydrolyzed Raw Cassava Starch ** | |
| --- | --- | --- | --- | --- | --- | --- |
| | Bioreactor | | | | | |
| | 7.5-L | 100-L | 7.5-L | 100-L | 5.0-L | 200-L |
| | Fermentation | | | | | |
| Ethanol (g·L$^{-1}$) | 73.2 ± 6.0 | 103.5 ± 2.3 | 79.6± 0.7 | 63.2 ± 1.1 | 81.9 ± 1.9 | 80.9 ± 0.5 |
| Ethanol Yield (%) | 71.2 ± 5.8 | 93.8 ± 1.8 | 99.5 ± 0.9 | 78.9 ± 0.4 | 75.3 ± 1.3 | 74.4 ± 0.3 |
| Sugar Consumption (%) | 81.5 ± 0.3 | 98.4 ± 0.1 | 92.2 ± 0.8 | 100.0 ± 0.0 | - | - |
| | By-products | | | | | |
| Glycerol (g·L$^{-1}$) | 4.8 ± 0.2 | 5.6 ± 0.1 | 8.7 ± 0.2 | 6.5 ± 0.2 | 11.4 ± 0.1 | 11.0 ± 0.0 |
| Lactic Acid (g·L$^{-1}$) | 5.7 ± 0.2 | 4.7 ± 0.6 | 3.8 ± 0.1 | 3.4. ± 0.2 | 0.5 ± 0.0 | 0.5 ± 0.0 |
| Acetic Acid (g·L$^{-1}$) | 0.0 ± 0.0 | 0.0 ± 0.0 | 0.8 ± 0.1 | 0.2 ± 0.0 | 0.0 ± 0.0 | 0.2 ± 0.0 |

Finally, if the results found in vials are compared with the 100 L bioreactor, it can be observed that they are very similar and, in that sense, the process should be scalable with reproducible results. On the other hand, in the case of the 7.5 L reactor, it is possible that the agitation speed of the system had a negative impact on the fermentation process, as we have already mentioned, and that the inhibition of fructose consumption decreased the ethanol yields.

## 4. Conclusions

The outcomes of the present work show the potential of cherry juice as a carbon source and whey as a nutrient source for bioethanol production using *Saccharomyces cerevisiae*. The urea addition improved the ethanol yield by up to 10%. Because urea is a cheap product, it is an excellent option that improves the yield and would not increase the costs of the processes. The sugar consumption (between 81.5% and 98.4%) during the fermentation process and the ethanol yield (between 71.2% and 93.8%), showed that a major part of the sugars were metabolized to ethanol under the operational conditions evaluated,. Other metabolites, such as glycerol and lactic acid, reached low values that were good for future distillation processes, and they demonstrated, once again, that the selected operational conditions are ideal for ethanol production using cherry juice supplemented with whey. Finally, cherry juice by itself is a medium that inhibits yeast growth, limiting yields in fermentation processes, and the addition of urea is not sufficient to diminish this toxic effect. Therefore, detoxification processes are necessary if it is to be used as the sole source of carbon for fermentation processes.

**Supplementary Materials:** The following supporting information can be downloaded at https://www.mdpi.com/article/10.3390/fermentation9020170/s1: Figure S1: Dry Cell Weight (DWC) versus Optical Density at 600 nm (OD600) of *Saccharomyces cerevisiae.* Table S1: Analysis of minerals present in whey.

**Author Contributions:** J.R.G.C.: conceptualization, methodology, investigation, formal analysis and writing—review and editing; J.-B.B.: conceptualization, methodology, investigation, formal analysis and writing—review and editing; J.M.d.M.D.: conceptualization, methodology, investigation, formal analysis and writing—review and editing; J.-M.L.: supervision, writing—review and editing, and funding acquisition. All authors have read and agreed to the published version of the manuscript.

**Funding:** This research was funded by the Conseil de Recherches en Sciences Naturelles et en Génie du Canada (ALLRP 555816–20) and the authors would also like to acknowledge the financial contribution from our industrial partners Gestion P.A.S and Laiterie de Coaticook Ltée.

**Institutional Review Board Statement:** Not applicable.

**Informed Consent Statement:** Not applicable.

**Data Availability Statement:** Not applicable.

**Acknowledgments:** The authors would like to thank Jean Provencher and Philippe Robert (Gestion P.A.S Inc.) for providing the industrial cherry juice, whey and all valuable information regarding their production. Further acknowledgement goes to the Analytical Chemistry Laboratory of the Biomass Technology Laboratory and to Thierry Ghislain and Maxime Lessard for their support in the analysis of the samples.

**Conflicts of Interest:** The authors declare no conflict of interest.

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
