# Peer review of "High-Gravity Fermentation for Bioethanol Production from Industrial Spent Black Cherry Brine Supplemented with Whey"

_fermentation, doi:10.3390/fermentation9020170_

Round 1

Reviewer 1 Report

Manuscript well-written and has novelty.

However, I have some points...

1. Please verify typos through the text, e.g. line 245 

2. The authors should add some table comparing their best results with some important data found in the literature.

Author Response

  • Reviewer #1

Manuscript well-written and has novelty.

Q1. Please verify typos through the text, e.g., line 245 

A1. Thank you for your recommendation. The revision was done and the line 245 was corrected.

Q2. The authors should add some table comparing their best results with some important data found in the literature.

A2. Thank for your observation. The table 2 was modified to make comparison among different substrates.

Reviewer 2 Report

I have listed some points needed the authors to explain.

Author Response

  • Reviewer #2

 In this work involves the production of bioethanol from industrial spend black cherry brine supplemented with whey via high gravity fermentation. The article is very interesting in several points such as raw material use as well as a technique for fermentation. After I have read throughout the whole manuscript. I do agree to accept this article to publish after the authors figure out in some points as following;

Q1. The authors should give more discussion about the raw materials used in both cherry juice and whey comparing to other raw materials. Since, they were quite expensive. A1. Thank you for your comment. Nonetheless, the substrates (Spend Cherry Brine and Whey) were free since they are by-products after the milk processing and ice cream production in our partner’s company. In any case we did the characterization of this by-products, we mentioned about their compositions in section 2.1 also we mentioned some characteristic in the introduction part.

Q2. Please check the Section 2.3 in line 229 should be changed to be the Section 3.2 (Figure 2) whether or not.

A2. Thank for your observation. No, the section we refer to is 2.3.

Q3. In Figure 2 (A), Please discuss more, when the CO2 production increased, the ethanol yields were also increased. For Figure 2 (B) the error bars for the cases of CJ20U0.0 and CJ20WU0.5 were quite broadly, why?

A3. Thank for your valuable input. (a) A CO2 discussion was added in the document. (b) In the case of the CJ20U0.0 the response was typical when a media is toxic for microorganisms some of them can start to metabolize some compounds but other just died show a variable behavior. In case the CJW20U0.5, the dilution in whey and urea addition increases the yeast response to fermentation process. In any case, the error bars were 5.3% and 4.2% respectively, which are acceptable error values for bioprocesses.

Q4. 2.3. Fermentation experiments. Line 162-163. Why were these reactor operating parameters assumed. Give an explanation in the text.

A4. Thank for your comment. These operating parameters were not optimized in this study. However, we already investigated these fermentation parameters in a previous study when optimizing the ethanol production from softwood residues. An explanatory sentence was added to the manuscript to clarify our choice (Line 171-173).

Q5. The authors should explain more or give the reason why the pretreatment process did not do.

A5. Thank you for highlighting this. In our case we try some pre-treatments such as active carbon and calcium carbonate to remove some yeast-toxic compounds. However, our goal was focusing on reducing the process cost reducing the number of steps in the fermentation. In the document we did not add this information because we wanted to present the cheapest process found.

Q6. For the Section of the conclusion, please make in only one passage.

A6. Thank you for your recommendation. The conclusion was done in only one passage.

Q7. In the Section of 3.2, lines 258-263 the authors made discussion that was contrast in the Section of conclusion especially the pretreatment process. Please clarify this.

A7. We mentioned that the addition of steps as pre-treatments increase the process cost. Our hypothesis was, just the addition of urea was enough to reach comparable results with those found with Park and Balinsky (ref (8)).

Q8. The authors should make comparison among a high gravity fermentation, normal fermentation and even a very high gravity fermentation in the Section of introduction or in the Section of results.

A8. Thanks for the recommendation, we think this was not the focus on this paper, since we want to find a combination of by-products to produce ethanol at low cost.

Reviewer 3 Report

The paper „High gravity fermentation for bioethanol production from industrial spend black cherry brine supplemented with whey” concerns the usage of two industrial by-products (spent cherry brine and whey) as a source of carbon and nutrients, respectively, for fermentation conducted by Saccharomyces cerevisiae yeast. This paper deals with a very relevant issue and also is well-written and interesting. I recommend the manuscript be accepted after a minor revision.

  1. The conditions that allowed for obtaining the best results should be included in the abstract.
  2. Have the Authors confirmed that 60 g/L of total sugars corresponds to 6°Bx? As far as I know, sugar represents approx 80%-90% of total soluble solid content
  3. L 86 – “The composition of the media used to evaluate yeast growth was cherry juice at 60 g·L-1 of total fermentable sugars (6.0°Brix) diluted in tap water and whey,…” – I suggest “tap water or whey” – right now it suggests that the juice was diluted with both 
  4. L119-120 - How was the agitation maintained 7.5L bioreactor?
  5. L143 and L146 – What was the concentration of the mobile phase? 
  6. L146 – Was the flow rate of the mobile phase indeed 0.08 mL/min? This is not a typical flow for the ion exclusion columns

Author Response

  • Reviewer #3

The paper „High gravity fermentation for bioethanol production from industrial spend black cherry brine supplemented with whey” concerns the usage of two industrial by-products (spent cherry brine and whey) as a source of carbon and nutrients, respectively, for fermentation conducted by Saccharomyces cerevisiae yeast. This paper deals with a very relevant issue and also is well-written and interesting. I recommend the manuscript be accepted after a minor revision.

Q1. The conditions that allowed for obtaining the best results should be included in the abstract.

A1. Thanks for your recommendation. The composition of the media and operational conditions were included in the abstract.

Q2. Have the Authors confirmed that 60 g/L of total sugars corresponds to 6°Bx? As far as I know, sugar represents approx 80%-90% of total soluble solid content.

A2. Yes, we did the sugar quantification of the concentrated cherry juice and whey by ion chromatography then the media were prepared base on initial fermentable sugars concentration in cherry juice.

Q3. L 86 – “The composition of the media used to evaluate yeast growth was cherry juice at 60 g·L-1 of total fermentable sugars (6.0°Brix) diluted in tap water and whey,” – I suggest “tap water or whey” – right now it suggests that the juice was diluted with both

A3. Thanks for highlight this. The correction was done. 

Q4. L119-120 - How was the agitation maintained 7.5L bioreactor?

A4. Thanks for highlight this. We added the information related to stirring system.

Q5. L143 and L146 – What was the concentration of the mobile phase? 

A5. Thanks for your question. We did the correction in the document. The concentration of the mobile phase was 2.50 mM H2SO4.

Q6. L146 – Was the flow rate of the mobile phase indeed 0.08 mL/min? This is not a typical flow for the ion exclusion columns.

A6. Thanks for your observation. The correction was done in the document, the flow rate of the mobile phase was 0.600 mL/min.
